# ^64^Cu-ATSM Predicts Efficacy of Carbon Ion Radiotherapy Associated with Cellular Antioxidant Capacity

**DOI:** 10.3390/cancers13246159

**Published:** 2021-12-07

**Authors:** Ankita Nachankar, Takahiro Oike, Hirofumi Hanaoka, Ayaka Kanai, Hiro Sato, Yukari Yoshida, Hideru Obinata, Makoto Sakai, Naoto Osu, Yuka Hirota, Akihisa Takahashi, Atsushi Shibata, Tatsuya Ohno

**Affiliations:** 1Department of Radiation Oncology, Gunma University Graduate School of Medicine, Maebashi 371-8511, Japan; drankitan@gmail.com (A.N.); hiro.sato@gunma-u.ac.jp (H.S.); m12201018@gunma-u.ac.jp (N.O.); yukahirota@gunma-u.ac.jp (Y.H.); tohno@gunma-u.ac.jp (T.O.); 2Gunma University Heavy Ion Medical Center, Maebashi 371-8511, Japan; yyukari@gunma-u.ac.jp (Y.Y.); sakai-m@gunma-u.ac.jp (M.S.); a-takahashi@gunma-u.ac.jp (A.T.); 3Department of Radiotheranostics, Gunma University Graduate School of Medicine, Maebashi 371-8511, Japan; hanaokah@hirakata.kmu.ac.jp (H.H.); kanai-a@gunma-u.ac.jp (A.K.); 4Laboratory for Analytical Instruments, Education and Research Support Center, Gunma University Graduate School of Medicine, Maebashi 371-8511, Japan; obi@gunma-u.ac.jp; 5Signal Transduction Program, Gunma University Initiative for Advanced Research (GIAR), Maebashi 371-8511, Japan; shibata.at@gunma-u.ac.jp

**Keywords:** carbon ion radiotherapy, relative biological effectiveness, ^64^Cu-ATSM, antioxidant systems, reactive oxygen species

## Abstract

**Simple Summary:**

Carbon ion radiotherapy is an emerging cancer treatment modality that has a greater therapeutic window than conventional photon radiotherapy. To maximize the efficacy of this extremely scarce medical resource, it is important to identify predictive biomarkers of higher carbon ion relative biological effectiveness (RBE) over photons. Here we show that the carbon ion RBE in human cancer cells correlates with the cellular uptake of ^64^Cu(II)-diacetyl-bis(N4-methylthiosemicarbazone) (^64^Cu-ATSM), a potential radioligand that reflects an over-reduced intracellular environment. High RBE/^64^Cu-ATSM cells show greater steady-state levels of antioxidant proteins and increased capacity to scavenge reactive oxygen species in response to X-rays than low RBE/^64^Cu-ATSM counterparts. These data suggest that the cellular antioxidant activity is a possible determinant of carbon ion RBE predictable by ^64^Cu-ATSM uptake.

**Abstract:**

Carbon ion radiotherapy is an emerging cancer treatment modality that has a greater therapeutic window than conventional photon radiotherapy. To maximize the efficacy of this extremely scarce medical resource, it is important to identify predictive biomarkers of higher carbon ion relative biological effectiveness (RBE) over photons. We addressed this issue by focusing on cellular antioxidant capacity and investigated ^64^Cu(II)-diacetyl-bis(N4-methylthiosemicarbazone) (^64^Cu-ATSM), a potential radioligand that reflects an over-reduced intracellular environment. We found that the carbon ion RBE correlated with ^64^Cu-ATSM uptake both in vitro and in vivo. High RBE/^64^Cu-ATSM cells showed greater steady-state levels of antioxidant proteins and increased capacity to scavenge reactive oxygen species in response to X-rays than low RBE/^64^Cu-ATSM counterparts; this upregulation of antioxidant systems was associated with downregulation of TCA cycle intermediates. Furthermore, inhibition of nuclear factor erythroid 2-related factor 2 (Nrf2) sensitized high RBE/^64^Cu-ATSM cells to X-rays, thereby reducing RBE values to levels comparable to those in low RBE/^64^Cu-ATSM cells. These data suggest that the cellular activity of Nrf2-driven antioxidant systems is a possible determinant of carbon ion RBE predictable by ^64^Cu-ATSM uptake. These new findings highlight the potential clinical utility of ^64^Cu-ATSM imaging to identify high RBE tumors that will benefit from carbon ion radiotherapy.

## 1. Introduction

Carbon ion radiotherapy (CIRT) is an emerging cancer treatment modality that has a greater therapeutic window than conventional photon radiotherapy [1]. Clinical evidence suggests that CIRT shows promising antitumor effects against a wide spectrum of cancers, including head and neck cancer [2], lung cancer [3,4], colorectal cancer [5], prostate cancer [6,7], and sarcoma [8]. Preclinical evidence suggests that carbon ions exert strong cell killing effects even against photon-resistant cells and that the relative biological effectiveness (RBE) of carbon ions over photons varies widely among different cell lines [9,10,11]. However, there is a limited number of CIRT facilities in operation worldwide [12]; this means that only <0.1% of newly diagnosed cancer patients have access to the modality [13]. From this standpoint, maximizing the use efficacy of CIRT by optimizing patient stratification is of great importance. To this end, the identification of biomarkers predictive of high RBE tumors is of utmost urgency.

DNA double-strand breaks (DSBs) are lethal events induced by ionizing radiation [14]. The direct and indirect effect is the predominant mechanism of DSB induction for high- and low-linear energy transfer (LET) radiations, respectively [14]. The indirect effect involves radiolysis of water, resulting in the production of reactive oxygen species (ROS), such as free radicals (H_2_O^+^, O_2_^−^, HO^−^); these ROS break the chemical bonds within DNA molecules to induce DNA damage [14]. Cellular antioxidant systems mitigate the indirect effects of ionizing radiation. Nuclear factor erythroid 2-related factor 2 (Nrf2) is a transcription factor and key regulator of cellular antioxidant systems; overexpression of Nrf2 upregulates expression of antioxidant genes. such as thioredoxin and glutathione [15,16]. Manganese superoxide dismutase (SOD2) catalyzes the dismutation of superoxide radicals into less reactive hydrogen peroxide molecules [17]. Evidence suggests that these antioxidant types of machinery are upregulated in a subset of cancers and that such cancers have an over-reduced intracellular environment.

^64^Cu(II)-diacetyl-bis(N4-methylthiosemicarbazone) (^64^Cu-ATSM) is a redox imaging tracer [18] with high lipophilicity and low redox potential; thus, ^64^Cu-ATSM diffuses passively across cell membranes and is reduced into a less soluble and unstable form [^64^Cu(I)-ATSM]^−^ after reacting with redox-active molecules, such as nicotinamide adenine dinucleotide phosphate or thiol groups [19]. Reduced ^64^Cu-ATSM undergoes protonation at the N3 and N6 positions to dissociate into copper, which is trapped irreversibly and accumulates in intracellular copper chaperone proteins and ATSM is conjugated with hydrogen molecules [18,20,21]. This redox-specific trapping mechanism leads to the accumulation of ^64^Cu-ATSM in over-reduced cells [18,19,20,21]. However, the association between the RBE of carbon ions and antioxidant capacity or ^64^Cu-ATSM uptake by cancer cells has not been elucidated.

In this study, we show for the first time that the RBE of carbon ions is associated with ^64^Cu-ATSM uptake and with antioxidant capacity in cancer cells. These new findings highlight the potential clinical utility of ^64^Cu-ATSM imaging for the identification of high RBE tumors that will benefit from CIRT.

## 2. Materials and Methods

### 2.1. Cells and Materials

A549, FaDu, H1299, H1650, H1703, HCT15, PC-3, and U2OS cells were obtained from ATCC (Manassas, VA, USA). PC9 was obtained from Riken Cell Bank (Tsukuba, Ibaraki, Japan). Ma24 was obtained from Dr. Shimizu (Tokushima University, Tokushima, Japan) [22]. The origin and histology are summarized in Appendix A. Cells were cultured at 37 °C/5% CO_2_ in RPMI-1640 (Sigma-Aldrich, St. Louis, MO, USA) supplemented with 10% fetal bovine serum (Life Technologies, Carlsbad, CA, USA). Brusatol, an Nrf2 inhibitor, was obtained from Sigma and prepared in dimethyl sulfoxide (Fujifilm Wako, Osaka, Japan).

### 2.2. Irradiation

X-ray irradiation of cultured cells was performed using MX-160Labo (160 kVp, 1.06 Gy/min; mediXtec, Matsudo, Japan). X-ray irradiation of mouse xenografts was performed using TITAN-225S (200 kVp, 1.30 Gy/min, Shimadzu, Otsu, Japan). CIRT was performed at Gunma University Heavy Ion Medical Center; cultured cells or mouse xenografts were irradiated at the center of a 6 cm-spread-out Bragg peak (290 MeV/nucleon, approximately 50 keV/μm).

### 2.3. Clonogenic Survival Assays

Clonogenic survival assays were performed as described previously with at least three technical and biological replicates [23,24]. Colonies comprising at least 50 cells were counted. The surviving fraction was normalized to that of the corresponding controls. The RBE of carbon ions over X-rays was calculated as D_50_ for X-rays/D_50_ for carbon ions, where D_50_ indicates the dose that provides 50% survival calculated by using the linear-quadratic model [25]. In the same manner, the sensitizer enhancement ratio (SER) of brusatol was calculated as D_50_ in the absence of brusatol/D_50_ in the presence of brusatol [26].

### 2.4. Immunoblotting

Immunoblotting of whole cell lysates was performed as described previously [27]. The intensity of a given protein band was quantified using ImageJ v1.48 (National Institutes of Health, Bethesda, MD, USA) and normalized to that of GAPDH (loading control). Uncropped images of the immunoblots are shown in Appendix A. Similar results were obtained in two independent experiments. The antibodies used in this study are listed in Appendix A.

### 2.5. ROS Assays

Cellular ROS production was assessed using a Cellular ROS Assay Kit (Abcam, Cambridge, UK). In this assay, 2′,7′-dichlorofluorescin diacetate (DCFDA), a non-fluorescent compound, is oxidized by ROS to produce fluorescent 2′,7′-dichlorofluorescin (DCF). Cells seeded in 6-well plates (2 × 10^5^ cells per well) were incubated at 37 °C for 48 h and then subjected to the treatment of interest. Cells were treated with DCFDA according to the manufacturer’s protocol and the DCF fluorescence signal was measured using an Attune NxT flow cytometer (Thermo Fisher Scientific, Waltham, MA, USA).

### 2.6. Assessment of TCA Cycle Intermediates

Steady-state cellular levels of TCA cycle intermediates were assessed using liquid chromatography coupled to a triple quadrupole mass spectrometer (LC-MS/MS; LC-MS-8050 system, Shimadzu). Cells seeded in 6-well plates (2 × 10^5^ cells per well) were incubated at 37 °C for 48 h, washed twice with phosphate-buffered saline, and then treated with 80% methanol for 30 min to extract metabolites. The extracts were purified using a Captiva™ ND*^Lipids^* filter plate (Agilent, Santa Clara, CA, USA), dried in a vacuum evaporator, resuspended in distilled water, and then subjected to LC-MS/MS analysis. The relative levels of metabolites were determined using the Method Package for Primary Metabolites (Shimadzu) and a Discovery HS F5-3 column (Sigma-Aldrich). The peak ion intensity of each metabolite was normalized to the sum of the peak ion intensity of all detected metabolites.

### 2.7. Assessment of Tumor Xenograft Growth

H1299 or HCT15 cells (5 × 10^6^ cells per mouse in 100 µL phosphate-buffered saline) were injected subcutaneously into the thigh of 6-week-old BALB/c female nude mice (Japan SLC, Hamamatsu, Shizuoka, Japan). When the tumors reached 100 mm^3^ (i.e., approximately 2–3 weeks after inoculation), the mice were stratified randomly into three groups: an X-ray group (10 Gy), a carbon ion group (10 Gy), and a sham-irradiated group. Tumor size and body weight were measured twice a week. The day of irradiation was defined as Day 0. Tumor volume (TV) was calculated using the formula: TV = (L × W^2^)/2, where L and W are the longest diameter and the perpendicular diameter of a tumor, respectively [28]. The antitumor effect of radiation was evaluated according to the time required for the TV to reach 400% of that measured on Day 0 (t_400%_) [29]. Based on the ethical standpoint, measurement was terminated when a mouse developed obvious weakness, skin metastasis, bleeding from tumor, or tumor exceeding 1000 mm^3^; these mice were euthanized in accordance with standard protocols. All animal experiments were approved by Gunma University Animal Experiment Committee (approval number: 18-016; approval date: 5 October 2018).

### 2.8. Assessment of ^64^Cu-ATSM Uptake

^64^Cu was produced using a biomedical cyclotron, CYPRIS HM-18 (Sumitomo Heavy Industries Ltd., Tokyo, Japan). ^64^Cu-ATSM was synthesized as described previously [30]. For in vitro assessment, 2 × 10^5^ cells seeded in 6-well plates were incubated overnight at 37 °C and then exposed to 10 kBq/mL of ^64^Cu-ATSM for 0, 30, or 60 min at 37 °C [31]. The cells were washed three times with cold phosphate-buffered saline and collected as a suspension in 1 mL of culture medium. The level of ^64^Cu-ATSM in the cell suspensions was measured using a gamma counter ARC7001 (Hitachi Aloka Medical, Tokyo, Japan). The measured ^64^Cu-ATSM levels were normalized to the number of cells after subtracting the background levels measured from the culture medium [31]. For the in vivo experiments, tumor xenografts were prepared as described in the previous section. ^64^Cu-ATSM (2 MBq per mouse) was administered intravenously. At 1 h post-administration, positron emission tomography (PET) was performed using an animal PET scanner (Inveon, Siemens, Knoxville, TN, USA), with an acquisition time of 5 min [31,32]. The imaging data were reconstructed using an iterative OSEM3D/MAP procedure, with a matrix size of 128 × 128 × 159 (including attenuation correction) and the maximum standardized uptake value (SUVmax) was calculated by placing the region of interest on the whole tumor using the Inveon Research Workplace workstation (Siemens).

### 2.9. Statistical Analysis

Differences between two groups were assessed using *t*-test and the Mann–Whitney U-test (for the dataset with and without a normal distribution, respectively); normality of a dataset was assessed using the Shapiro–Wilk test. The correlation between two datasets was assessed using Spearman’s rank test. The survival of mice was analyzed using the Kaplan–Meier method and the log rank test. The ability of SUVmax to predict the cell line of xenograft was examined by receiver operating characteristic (ROC) analysis. All statistical tests were two-tailed. A *p*-value < 0.05 (after Bonferroni correction in the case of multiple testing) was considered statistically significant. Survival analysis and ROC analysis were performed using SPSS Statistics 27 (SPSS Inc., Chicago, IL, USA). All other statistical analyses were performed using GraphPad Prism 8 (GraphPad Software Inc., San Diego, CA, USA).

## 3. Results

### 3.1. Carbon Ion RBE Correlates with ^64^Cu-ATSM Uptake In Vitro

To investigate the association between the RBE of carbon ions and cellular ^64^Cu-ATSM uptake in cancer cells in vitro, we first assessed the X-ray- or carbon ion-sensitivity of ten human cancer cell lines originated from the cancer types targeted by CIRT in the clinic (Appendix A) [2,3,4,5,6,7,8]. The cell lines showed various sensitivities to the radiation; however, the cell killing effect was greater for carbon ions than for X-rays in all cell lines examined (Figure 1A). The resultant RBE values ranged from 1.3 ± 0.15 (for HCT15) to 2.8 ± 0.48 (for H1299), with a median of 2.0 (Figure 1B); this is consistent with the historical context of the clinical beam set-up of CIRT, for which the HSG cell line (with an RBE of ~2.0) was chosen as the reference because it shows an intermediate RBE [33]. Together, these data suggest the validity of this cell line panel for use in this study.

Next, we assessed cellular ^64^Cu-ATSM uptake by the cancer cell lines under normoxic conditions (Figure 1C). Uptake peaked at 30 min, as described previously [20,31,34,35], and cells were saturated at 60 min post-treatment; thus, we used data obtained at the 30 min point for correlation analysis. RBE showed strong a correlation with ^64^Cu-ATSM uptake (Figure 1D) suggesting that the RBE for carbon ions correlates with ^64^Cu-ATSM uptake in cancer cells in vitro.

### 3.2. The Carbon Ion RBE Correlates with ^64^Cu-ATSM Uptake In Vivo

To validate these findings in an in vivo setting, we investigated the correlation between RBE and ^64^Cu-ATSM uptake in a nude mouse xenograft model. H1299 and HCT15 cells, which showed the highest and lowest RBE, respectively in vitro, were chosen for these experiments (Figure 1D). In H1299 xenografts, carbon ions had a significantly greater growth suppression effect than X-rays at the same physical dose (Figure 2A,B). Consistent with this, H1299 xenograft-bearing mice in the carbon ion-treated group survived for significantly longer than mice in the X-ray-treated group (Figure 2C). By contrast, the growth suppression effect in the HCT15 xenograft group was comparable between carbon ions and X-rays (Figure 2A,B), as was the survival of HCT15 xenograft-bearing mice (Figure 2C). There were no significant inter-group differences in body weight (data not shown). Taken together, the data show that the relative antitumor effect of carbon ions to X-rays observed in the xenograft model was consistent with in vitro RBE, suggesting the robustness of the xenograft models as a means of in vivo validation. Notably, ^64^Cu-ATSM uptake by H1299 xenografts was significantly greater than that by HCT15 xenografts (Figure 2D). The SUVmax for ^64^Cu-ATSM uptake predicted the cell line for xenografts significantly (Appendix A). Taken together, the in vitro and in vivo data strongly suggest that carbon ion RBE correlates with ^64^Cu-ATSM uptake by cancer cells.

### 3.3. Upregulation of Antioxidant Systems Plays a Role in the High Carbon Ion RBE and ^64^Cu-ATSM Uptake

Having found a correlation between the RBE and ^64^Cu-ATSM uptake, we sought to explore the underlying mechanisms. Based on evidence suggesting that ^64^Cu-ATSM accumulates in over-reduced cells [18], and that induction of DSBs by X-rays relies heavily on the indirect effects induced via intracellular ROS [14], we focused on the antioxidant capacity of cancer cells. Interestingly, cells with a high RBE and high ^64^Cu-ATSM uptake (i.e., A549 and H1299 cells) showed higher expression of antioxidant proteins (i.e., Nfr2, SOD2, and TRX1) under steady-state conditions than cells with a low RBE and low ^64^Cu-ATSM uptake (i.e., H1650 and HCT15 cells) (Figure 3A). In addition, high RBE/^64^Cu-ATSM cells showed greater ROS scavenging capacity in response to X-ray irradiation than low RBE/^64^Cu-ATSM cells (Figure 3B). Furthermore, high RBE/^64^Cu-ATSM cells showed lower levels of TCA cycle intermediates than low RBE/^64^Cu-ATSM cells under steady-state conditions (Figure 3C); this is reasonable because Nrf2-driven upregulation of cellular antioxidant systems causes increased glutamate consumption, leading to downregulation of mitochondrial respiration [36,37]. Together, these data suggest that the upregulation of antioxidant systems is the potential mechanism underlying high carbon ion RBE and ^64^Cu-ATSM uptake by cancer cells.

To further consolidate the above findings, we investigated the effect of Nrf2 inhibition on sensitivity to X-rays or carbon ions since Nrf2 is a major regulator of cellular antioxidant activity [38]. Brusatol, an Nrf2 inhibitor, efficiently suppressed the expression of Nrf2 and associated antioxidant proteins in high RBE/^64^Cu-ATSM cells in a concentration-dependent manner (Figure 4A,B). Brusatol (50 nM) was used for the radiosensitization experiments because it showed mild toxicity when used as a single agent (Appendix A). Notably, brusatol sensitized high RBE/^64^Cu-ATSM cells to X-rays (Figure 4C,D). By contrast, this X-ray-sensitizing effect was not observed in low RBE/^64^Cu-ATSM cells (Figure 4C,D). The sensitizing effect of brusatol was not observed for carbon ions, irrespective of RBE/^64^Cu-ATSM uptake (Figure 4C, D). As a result, the brusatol-modified RBE values for the high and low RBE/^64^Cu-ATSM cells were comparably low (Figure 4C). Taken together, these data suggest that the activity of antioxidant systems, driven by Nrf2, is a potential determinant of carbon ion RBE, which can be predicted by measuring ^64^Cu-ATSM uptake.

## 4. Discussion

In the clinic, CIRT is delivered in a dose measured in the unit Gy (RBE), which is determined according to the RBE value of a human salivary gland cell line (HSG); this was approximately 2.0 for the beam settings used in this study [25]. However, RBE values among individual cancers vary widely, even among cancers with the same histological, genetic, or virus-infectious status [9,10,11]. From this standpoint, greater antitumor effects, as well as a wider therapeutic window, are anticipated for tumors showing RBE values greater than that of HSG cells. Therefore, such tumors should be stratified preferentially for CIRT. In this study, we found that cellular antioxidant activity is a potential determinant of carbon ion RBE. The data indicate that high RBE tumors can be identified by PET imaging using ^64^Cu-ATSM, a finding that warrants clinical validation. In addition, we found that the high RBE/^64^Cu-ATSM uptake status of cancer cells was associated with the downregulation of TCA cycle intermediates. These data indicate the potential of the TCA cycle activity as a surrogate biomarker of high RBE/^64^Cu-ATSM uptake. This should be pursued further since these metabolites can be assessed by LC-MS (which is more inexpensive than PET imaging).

^64^Cu-ATSM was originally developed as a marker of hypoxia [39,40,41]. Nevertheless, accumulating evidence suggests that this compound does not fully reflect the intratumor hypoxic fraction; this is supported by the fact that the intratumor spatial distribution of ^64^Cu-ATSM does not fully match that of other hypoxia markers [42,43]. Recent studies highlight another emerging biological property of ^64^Cu-ATSM: that it represents an intracellular over-reduced state [18,32,44]. This intracellular over-reduced status overlaps, at least partially, with hypoxia. Intratumoral accumulation of ^64^Cu-ATSM in xenograft model is a complex phenomenon influenced by various biological factors including tumor size, hypoxia fraction, tumor vascularity, and perfusion as discussed in previous publications [18,32,33]. Thus, in our in vivo experiments, we intended to minimize these variances by performing all experiments under normoxic conditions and by assessing tumor xenograft growth at a range of approximately 100–500 mm^3^ (considering the greater probability of the presence of hypoxic regions in larger tumors containing a necrotic center). Nevertheless, we found a smaller difference in ^64^Cu-ATSM uptake between the two groups compared with that observed in cell-based experiments, indicating the difficulty in identifying putative high-RBE tumors by ^64^Cu-ATSM PET in the clinic. Since human tumors do harbor similar complex biological contexts, this point must be further pursued toward clinical application.

Brusatol did not sensitize tumors to carbon ions, irrespective of the RBE and ^64^Cu-ATSM uptake. This is consistent with the understanding that direct ionization is the predominant mode of DSB induction by carbon ions [14,45,46,47]. More importantly, in the presence of brusatol, RBE values for high and low RBE/^64^Cu-ATSM cells were comparably low (i.e., <2.0). These data suggest that the RBE of carbon ions is dependent largely on the cellular capacity to mitigate the indirect effect of X-rays, whereas the cell killing effect of carbon ions against various cancers is more consistent than that of X-rays, supporting the rationale that photon-resistant tumors should be treated with CIRT.

The study has the following limitations. First, we did not use LET-specific carbon ion beams. This was because we intended to mimic the clinical situation by using SOBP beams that have mixed LET profiles. Second, in the LC-MS analysis, we did not analyze the metabolites other than the six TCA cycle intermediates (Figure 3C) due to technical difficulties. A more comprehensive analysis, taking these factors into account, will provide more detailed mechanistic insight into the association between carbon ion RBE, antioxidant activity, and ^64^Cu-ATSM uptake by cancer cells.

## 5. Conclusions

In the present study, we show for the first time that the RBE of carbon ions is associated with ^64^Cu-ATSM uptake and with antioxidant capacity in cancer cells. These new findings highlight the potential utility of ^64^Cu-ATSM imaging to identify high RBE tumors that will benefit from CIRT.

## Figures and Tables

**Figure 1 cancers-13-06159-f001:**
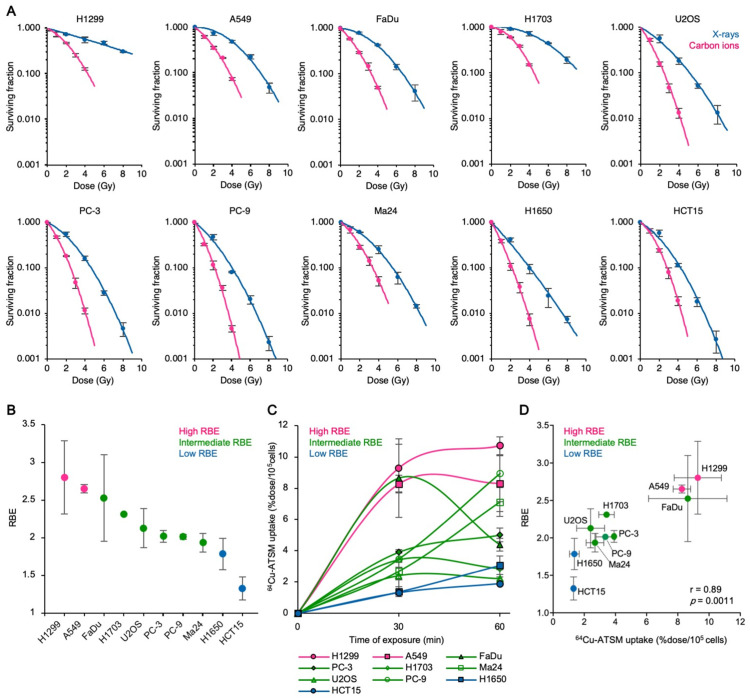
The carbon ion RBE correlates with ^64^Cu-ATSM uptake in vitro. (**A**) Sensitivity of various human cancer cell lines to X-rays or to carbon ions, as assessed in clonogenic assays (mean ± SD; *n* = 3). (**B**) RBE of carbon ions at D_50_ (i.e., the dose that provides 50% survival) was calculated from the data in A (mean ± SD; *n* = 3). (**C**) Intracellular ^64^Cu-ATSM uptake by cancer cell lines (mean ± SD; *n* = 3). (**D**) Correlation between RBE and ^64^Cu-ATSM uptake (mean ± SD; *n* = 3). Data presented in (**B**,**C**) (30 min) were used for correlation analysis. The *p* values and correlation coefficients (*r*) calculated using Spearman’s rank test are shown.

**Figure 2 cancers-13-06159-f002:**
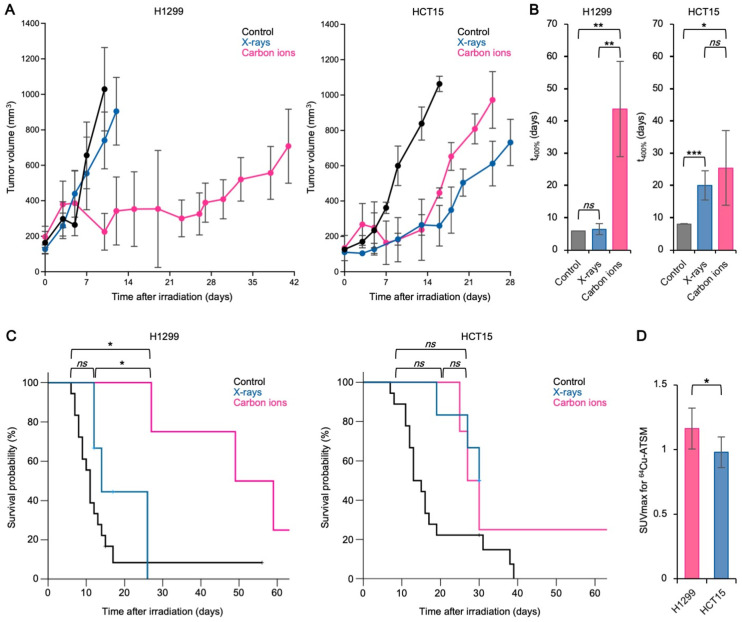
The carbon ion RBE correlates with ^64^Cu-ATSM uptake in vivo. (**A**) Growth of tumor xenografts treated with carbon ions (10 Gy) or X-rays (10 Gy) (mean ± SD; *n* = 4). (**B**) t_400%_ (i.e., the time required for the tumor volume to reach 400% of that measured on Day 0) for the data presented in A. (**C**) Kaplan-Meier survival estimates of mice bearing HCT15 or H1299 xenografts (*n* = 20, four, and four mice in the control, X-ray, and carbon ion groups, respectively). (**D**) ^64^Cu-ATSM uptake by tumor xenografts, as assessed by PET imaging (mean ± SD; *n* = 9). * *p* < 0.05; ** *p* < 0.01; *** *p* < 0.001. *ns*, not statistically significant.

**Figure 3 cancers-13-06159-f003:**
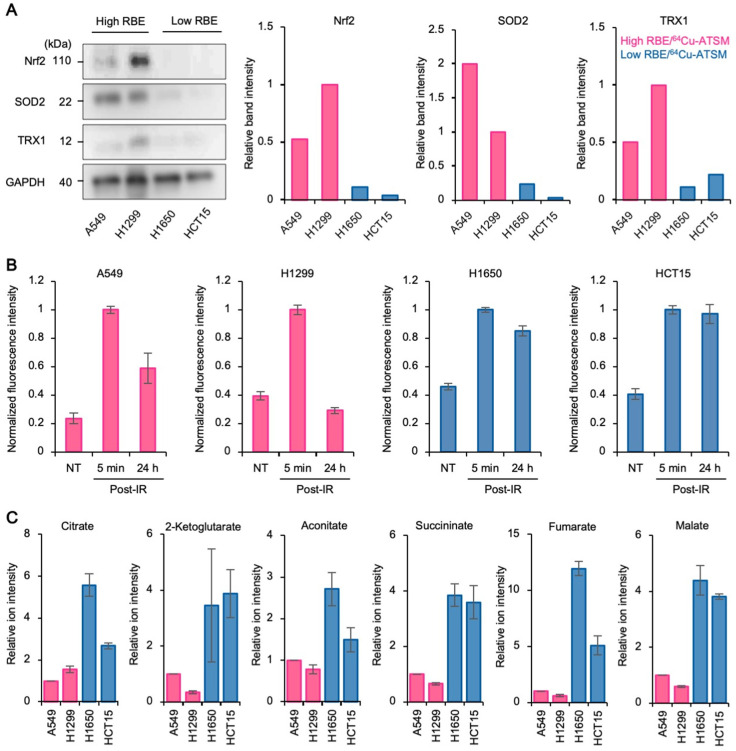
Upregulation of antioxidant systems plays a role in high RBE/^64^Cu-ATSM uptake. (**A**) Greater expression of antioxidant proteins in high RBE/^64^Cu-ATSM cells than in low RBE/^64^Cu-ATSM cells under steady-state conditions, as assessed by immunoblot analysis. Bar graphs show quantitated band intensities, normalized according to GAPDH. (**B**) High RBE/^64^Cu-ATSM cells show greater ROS scavenging capacity in response to X-ray irradiation than low RBE/^64^Cu-ATSM cells. Cells were treated with X-rays (4 Gy) and assessed using the Cellular ROS Assay (Abcam). For each cell line, DCF fluorescence signals, as measured by flow cytometry, are shown after normalizing to those obtained 5 min post-irradiation (mean ± SD; *n* = 3). (**C**) Downregulation of TCA cycle intermediates in high RBE/^64^Cu-ATSM uptake cells compared to low RBE/^64^Cu-ATSM uptake cells, as assessed by LC-MS/MS (mean ± SD; *n* = 3).

**Figure 4 cancers-13-06159-f004:**
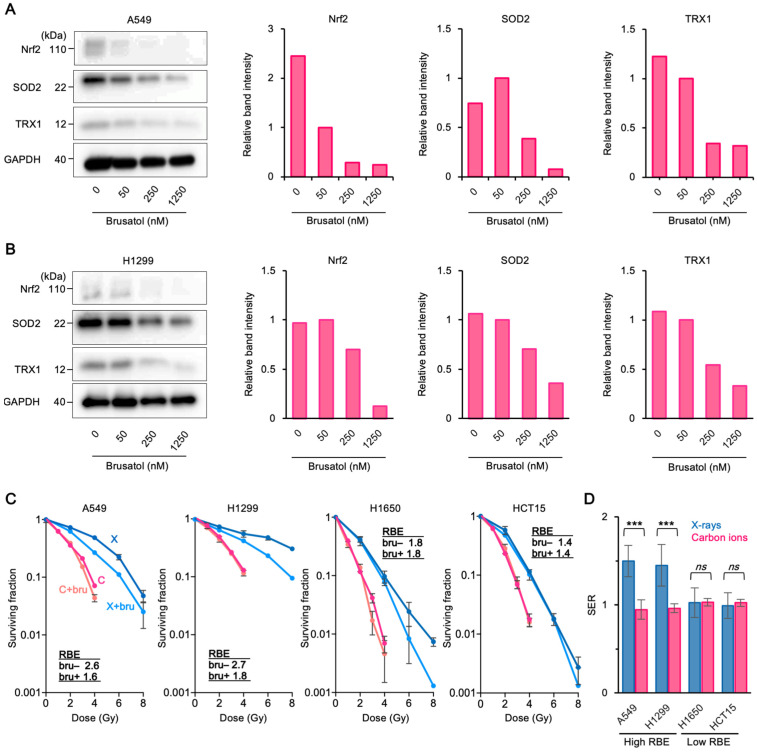
Nrf2 inhibition sensitizes high RBE/^64^Cu-ATSM cells to X-rays and lowers the RBE. (**A**,**B**) Brusatol suppresses the expression of antioxidant proteins in A549 cells (**A**) or H1299 cells (**B**) in a concentration-dependent manner. Cells were exposed to brusatol for 1 h and protein expression was assessed by immunoblotting. Bar graphs show quantitated band intensities, normalized according to GAPDH. (**C**) Sensitivity of high RBE/^64^Cu-ATSM cells (A549 and H1299) or low RBE/^64^Cu-ATSM cells (H1650 and HCT15) to X-rays or carbon ions in the presence or absence of brusatol, as assessed in clonogenic assays (mean ± SD; *n* = 3). Cells were exposed to brusatol (50 nM) or vehicle from 1 h pre-irradiation to the day of colony staining. C, carbon ions; X, X-rays; bru, brusatol. (**D**) Sensitizer enhancement ratio (SER: i.e., the ratio of the dose providing 50% survival in the absence of brusatol to that in the presence of brusatol) calculated from data presented in (**C**). *** *p* < 0.001. *ns*, not statistically significant.

## Data Availability

The data presented in the current study are available from the corresponding author (T.O. (Takahiro Oike)) upon reasonable request.

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
