# Peer review of "64Cu-ATSM Predicts Efficacy of Carbon Ion Radiotherapy Associated with Cellular Antioxidant Capacity"

_cancers, 2021, doi:10.3390/cancers13246159_

Round 1

Reviewer 1 Report

The manuscript describes that carbon ion relative biological effectiveness is correlated with 64Cu-ATSM uptakes, as evidenced in human cell lines and xenograft animal models. In addition, the author found that cellular antioxidant activity is associated with carbon ion relative biological effectiveness, high BRE tumor can be validated by 64Cu-ASTM PET imaging. Furthermore, the authors found that the TCA cycle activity could serve as a surrogate biomarker of high BRE and 64Cu-ATSM uptake. The exciting study may identity high BRE tumors that are resistant to X-ray therapy with 64Cu-ATSM PET imaging. The study is appropriately designed, and the data well support the conclusions. I recommend the manuscript be published in Cancers after the authors address the following issues:

  1. In figure 2A, xenograft tumor mice with two cell lines are monitored for different days, particularly for HCT15 cells, carbon ions led to faster tumor growth than X-rays. Could you please explain?
  2. The manuscript mentioned several times that 64Cu-ATSM is a potential biomarker of an over-reduced intracellular environment. 64Cu-ATSM should be a radioligand to detect the biomarker instead of a biomarker
  3. Lien 76, “redox imaging tracer molecule” may be changed to “redox imaging tracer”; Please give a full name of NADPH; line 143, “100 phosphate-buffered saline” may be changed to “100 uL phosphate-buffered saline”.

Author Response

Editor

Kind note: Please add markers or scale bars for all WB figures.

Response:

We sincerely thank the editor for evaluating our manuscript and for the comment. According to the suggestion, scales were provided to all Western blot images (Supplementary Figures S1 and S2).

Reviewer #1

The manuscript describes that carbon ion relative biological effectiveness is correlated with 64Cu-ATSM uptakes, as evidenced in human cell lines and xenograft animal models. In addition, the author found that cellular antioxidant activity is associated with carbon ion relative biological effectiveness, high BRE tumor can be validated by 64Cu-ASTM PET imaging. Furthermore, the authors found that the TCA cycle activity could serve as a surrogate biomarker of high BRE and 64Cu-ATSM uptake. The exciting study may identity high BRE tumors that are resistant to X-ray therapy with 64Cu-ATSM PET imaging. The study is appropriately designed, and the data well support the conclusions. I recommend the manuscript be published in Cancers after the authors address the following issues.

Response:

We sincerely thank the reviewer for evaluating our manuscript and for encouraging comments. According to the suggestions, the manuscript was revised as follows.

  1. In figure 2A, xenograft tumor mice with two cell lines are monitored for different days, particularly for HCT15 cells, carbon ions led to faster tumor growth than X-rays. Could you please explain?

Response:

We sincerely thank the reviewer for the important comment. The reason for the different days of monitoring was that we terminated the experiments by following the institutional ethical standard on animal experiments: i.e., the experiments should be terminated when a mouse developed obvious weakness, skin metastasis, bleeding from tumor, or tumor exceeding 1,000 mm3. This was clarified in Materials and Methods section (lines 153155).

Pertaining to the growth of HCT15 tumor xenografts, the difference between carbon ions and X-rays was statistically insignificant (P = 0.29, indicated as ns in Figure 2B) evaluated by using t400% as the endpoint (Jorgensen et al. Cancer Chemother Pharmacol 2007;59:725–732). Although we agree with the reviewer's impression that carbon ion-treated tumors looked to grow faster than X-ray-treated counterparts, we would like to note that, in this experimental systems, tumor growth kinetics become less reliable (i.e., less reproducible) after tumor volume reached approximately 500–1000 mm3; this is due to random development of intratumoral hypoxia-induced necrosis. That is why we chose t400% as the endpoint for tumor growth. Meanwhile, all the measurement data obtained until termination were honestly presented in the figures. Taken together, we believe that the data presented in Figure 2 is robust.

  1. The manuscript mentioned several times that 64Cu-ATSM is a potential biomarker of an over-reduced intracellular environment. 64Cu-ATSM should be a radioligand to detect the biomarker instead of a biomarker.

Response:

We thank the reviewer for the critical comment that we agree with. According to the suggestion, the description was changed to "64Cu-ATSM is a potential radioligand that reflects an over-reduced intracellular environment" (lines 26 and 36).

  1. Line 76, “redox imaging tracer molecule” may be changed to “redox imaging tracer”.

Response:

We thank the reviewer for the comment. The revision was made according to the suggestion.

Please give a full name of NADPH.

Response:

We thank the reviewer for the comment. NADPH was spelled out as nicotinamide adenine dinucleotide phosphate in lines 7980.

line 143, “100 phosphate-buffered saline” may be changed to “100 uL phosphate-buffered saline”.

Response:

Yes, here we meant "100 uL" as speculated. We apologize for the typo. The revision was made accordingly (line 144). We thank the reviewer for the comment.

Reviewer 2 Report

Dear authors,

Your manuscript on the use of 64Cu-ATSM for prediction of CIRT efficacy presents an interesting case study on the in vitro and in vivo correlation between uptake of the compound and tumor cell killing / RBE. I would like to congratulate you with an interesting study, and have only a few minor comments:

- In your in vitro study, the link between 64Cu uptake and high RBE is clearly there. However, in the in vivo study (figure 2) where you also looked at the uptake in the tumor xenografts, the difference in 64Cu uptake between the two cell lines is less convincing. Although your statistical test indicates there is a significant difference between the two cell lines (fig 2D), the large overlap in error bars suggests that in vivo it might be very difficult to actually assess whether a tumor can benefit from CIRT treatment based on the PET Images. I would like to suggest to add representative PET images of the mice to show the actual difference in uptake, and also to add in your discussion the feasibility of this method in vivocompared to the in vitro results.

- line 143: the 100 PBS, I’m assuming its microliters?

- line 161 kBq is written with a small k, not capital K. 

Author Response

Editor

Kind note: Please add markers or scale bars for all WB figures.

Response:

We sincerely thank the editor for evaluating our manuscript and for the comment. According to the suggestion, scales were provided to all Western blot images (Supplementary Figures S1 and S2).

Reviewer #2

Your manuscript on the use of 64Cu-ATSM for prediction of CIRT efficacy presents an interesting case study on the in vitro and in vivo correlation between uptake of the compound and tumor cell killing / RBE. I would like to congratulate you with an interesting study, and have only a few minor comments.

Response:

We sincerely thank the reviewer for evaluating our manuscript and for encouraging comments. According to the suggestion, the manuscript was revised as follows.

In your in vitro study, the link between 64Cu uptake and high RBE is clearly there. However, in the in vivo study (figure 2) where you also looked at the uptake in the tumor xenografts, the difference in 64Cu uptake between the two cell lines is less convincing. Although your statistical test indicates there is a significant difference between the two cell lines (fig 2D), the large overlap in error bars suggests that in vivo it might be very difficult to actually assess whether a tumor can benefit from CIRT treatment based on the PET Images. I would like to suggest to add representative PET images of the mice to show the actual difference in uptake, and also to add in your discussion the feasibility of this method in vivo compared to the in vitro results.

Response:

We sincerely thank the reviewer for the critical comments. We agree with the reviewer that the error bars for the data presented in Figure 2D have large overlap. In fact, the difference in 64Cu-ATSM uptake between two groups in xenograft experiments was obviously smaller than that observed in cell-based experiments, although it was statistically significant. In line with this, the PET images are not impressive (please see the Figure R1 below). Under such circumstances, we think that the addition of representative images (i.e., champion data) is inappropriate because it may lead to manipulation of the reader's impression.

Instead of not presenting the PET images, we added the detailed discussion on this point as follows: "Intratumoral accumulation of 64Cu-ATSM in xenograft model is a complex phenomenon influenced by various biological factors including tumor size, hypoxia fraction, tumor vascularity, and perfusion as discussed in previous studies (Colombié et al. Front Med 2015;2:58, Burgman et al. Nucl Med Biol 2005;32:623–630, Floberg et al. J Nucl Med 2020;61:427–432). Thus, in our in vivo experiments, we intended to minimize these variances by performing all experiments under normoxic conditions and by assessing tumor xenograft growth at a range of approximately 100–500 mm3 (considering the greater probability of the presence of hypoxic regions in larger tumors containing a necrotic center). Nevertheless, we found obviously smaller difference in 64Cu-ATSM uptake between two groups compared with that observed in cell-based experiments, indicating the difficulty in identifying putative high-RBE tumors by 64Cu-ATSM PET in the clinic. Since human tumors do harbor similar complex biological context, this point must be further pursued toward clinical application" (lines 313324). Again, we sincerely thank the reviewer for this comment and hope for your understanding.

Figure R1. Representative 64Cu-ATSM PET images. (A) High-RBE H1299 tumor xenograft. (B) Low-RBE HCT15 tumor xenograft.

line 143: the 100 PBS, I’m assuming its microliters?

Response:

We apologize for the typo. Yes, here we meant "100 uL" as speculated. The revision was made accordingly. We thank the reviewer for the comment.

line 161: kBq is written with a small k, not capital K.

Response:

We apologize for the typo. The revision was made accordingly. We thank the reviewer for the comment.
